# Association of Twelve Candidate Gene Polymorphisms with the Intramuscular Fat Content and Average Backfat Thickness of Chinese Suhuai Pigs

**DOI:** 10.3390/ani9110858

**Published:** 2019-10-23

**Authors:** Binbin Wang, Pinghua Li, Wuduo Zhou, Chen Gao, Hang Liu, Huixia Li, Peipei Niu, Zongping Zhang, Qiang Li, Juan Zhou, Ruihua Huang

**Affiliations:** 1Institute of Swine Science, Nanjing Agricultural University, Nanjing 210095, China; 2018205003@njau.edu.cn (B.W.); zhouwuduo@njau.edu.cn (W.Z.); GC1594140795@126.com (C.G.); lhomme@126.com (H.L.); lihuixia@njau.edu.cn (H.L.); 2Huaian Academy, Nanjing Agricultural University, Huaian 223005, China; niupeipei2@126.com (P.N.); zongpingzhang@126.com (Z.Z.); 3Huaiyin Pig Breeding Farm of Huaian City, Huaian 223322, China; 13912078083@139.com (Q.L.); 15261762741@163.com (J.Z.)

**Keywords:** Suhuai pig, intramuscular fat content, average backfat thickness, candidate gene

## Abstract

**Simple Summary:**

Appropriate intramuscular fat content (IFC) is the goal of consumers and the direction that breeders must pursue. However, it is difficult to improve the IFC but not average backfat thickness (ABT) by traditional breeding methods, and pigs must be slaughtered to accurately measure IFC. Marker-assisted selection (MAS) provides an economic and efficient method to improve the IFC in pigs. Our research indicated that the *FABP3* (rs1110770079) single nucleotide polymorphism (SNP) could be a candidate gene associated with IFC (but not ABT), and IFC could be improved by selecting the individuals with a favorable genotype (GG) of *FABP3* (rs1110770079) SNP for pig breeding.

**Abstract:**

The present study aimed to identify the molecular markers for genes that influence intramuscular fat content (IFC), but not average backfat thickness (ABT). A total of 330 Suhuai pigs were slaughtered, and measurements of IFC and ABT were obtained. Phenotypic and genetic correlations between IFC and ABT were calculated. Thirteen single nucleotide polymorphisms (SNPs) among 12 candidate genes for IFC were analyzed, including *FABP3*, *LIPE*, *IGF1*, *IGF2*, *LEP*, *LEPR*, *MC4R*, *PHKG1*, *RETN*, *RYR1*, *SCD,* and *UBE3C*. Associations of the evaluated SNPs with IFCIFC and ABT were performed. Our results showed that the means of IFC and ABT were 1.99 ± 0.03 % and 26.68 ± 0.28 mm, respectively. The coefficients of variation (CVs) of IFC and ABT were 31.21% and 19.36%, respectively. The phenotypic and genetic correlations between IFC and ABT were moderate. Only the *FABP3* (rs1110770079) was associated with IFC (*p* < 0.05) but not with ABT. Besides, there was a tendency for associations of *RYR1* (rs344435545) and *SCD* (rs80912566) with IFC (*p* < 0.1). Our results indicated that the *FABP3* (rs1110770079) SNP could be used as a marker to improve IFC without changing ABT in the Suhuai pig breeding system.

## 1. Introduction

Meat is widely consumed as an important protein source for humans; pork occupies a dominant position in the consumption structure of meat in China [1], and its consumption is about 40 kg per person per year in recent years. During the past decades, breeders have mainly focused on the increase of growth rate, muscle deposition, and lean meat yield, resulting in a reduction in meat quality and fat deposition [2,3]. Intramuscular fat content (IFC) is a key indicator of the meat quality assessment system. Previous studies have indicated that IFC is closely related to flavor, juiciness, tenderness, and water-holding capacity [4,5]. IFC refers to the amount of fat accumulated between muscle fibers or within muscle cells, which could be extracted from muscle samples by using chemical methods [6]. The main components of IFC include phospholipids and triglycerides [7]. Recently, some studies have shown that the increase in triglycerides contributes to increases in IFC [8,9].

At present, improving IFC has become an important objective in the modern pig breeding program [10], especially in China [11]. However, it is difficult to improve IFC by traditional breeding methods. Although the heritability of IFC is relatively high (h^2^ = 0.25–0.50) [12], it is still impossible to assess exact IFC in the offspring. Accurate measurement of IFC still requires slaughtering pigs, which is costly to implement. As compared to traditional breeding methods, marker-assisted selection (MAS) and genome selection (GS) technology are ideal ways to improve IFC in pigs. However, in comparison with MAS, many pig farms cannot cover the cost of GS technology. Thus, MAS is an attractive approach to improve IFC in pigs. Several candidate genes and causative mutations for IFC have been identified in previous studies, such as *FABP3* (rs1110770079) [13], *LIPE* (rs328830166) [14], *IGF1* (rs322131043), *IGF1* (rs341412920) [15], *IGF2* (g.3072G > A) [16], *LEP* (rs45431504) [17], *LEPR* (rs45435518) [18], *MC4R* (rs81219178) [19], *PHKG1* (rs697732005) [20], *RETN* (rs327132149) [21], *RYR1* (rs344435545) [22], *SCD* (rs80912566) [18] and *UBE3C* (rs81329544) [23].

In general, the phenotypic correlation between backfat (BF) thickness and IFC is moderate and positive. Therefore, increasing IFC often leads to increased BF thickness [24]. The increased BF thickness in turn could result in a decrease in lean meat percentage, which adversely affects the lean meat yield. Thus, it is especially necessary to find SNPs that could increase IFC but not the BF thickness. 

Suhuai pig is a new lean-type pig breed. Briefly, after 23 years of successive breeding of the crossbred offspring of the Large White Pig and Huai Pig, a new breed was developed, called the Xinhuai Pig, which contains 50% Large White and 50% Huai pig (1954–1977). Subsequently, Large White pigs were again used to cross with Xinhuai pigs, and their offspring were selected and bred for 12 years to obtain a new national lean type breed called the Suhuai pig, which contains 25% Chinese Huai pig and 75% Large White (1998–2010) (Figure 1). At present, Suhuai pigs are distributed in over 20 provinces such as Jiangsu and Anhui. There are more than 10,000 Suhuai sows, which produce more than 12,000 tons of pork per year. The lean meat percentage of carcass is about 57%, and the average daily gain from 30 kg to 90 kg is about 660 g. Huai pig is one of the North Chinese pig breeds with the characteristics of Chinese local pig such as high-meat quality and high-forage tolerance [25]. Historical data showed that Huai pig has relatively high IFC [26]. In contrast, Large White pig, a commercial breed that is known for its lean carcass and rapid growth rate, shows lower fat deposition capacity [27]. Therefore, as a synthetic crossbreed, Suhuai pig has the high forage tolerance of the Chinese Huai pig, as well as the rapid growth rate and high lean meat yield of the Large White breed. 

IFC is affected by many factors, such as heredity, nutrition, environment, and feeding methods. A previous study has shown that the CV of IFC was large within Suhuai pigs [28]. In the present study, all of the Suhuai pigs were raised under commercial conditions using the same feed. Considering that the heritability of IFC is relatively high, we speculate that the variation of IFC of Suhuai pigs is mainly due to genetic factors. Based on this, Suhuai pig is a suitable genetic material for identifying the molecular markers for IFC. The present study aimed to explore the association of the above-mentioned 13 SNPs with IFC and ABT of Suhuai pigs, and to find molecular markers that indicate higher IFC but not ABT, which will contribute to early selection of replacement pigs.

## 2. Materials and Methods

### 2.1. Ethics Statement

All experimental animals were raised according to Guidelines for the Care and Use of Laboratory Animals prepared by the Institutional Animal Welfare and Ethics Committee of Nanjing Agricultural University, Nanjing, China. All experimental protocols were approved by the Nanjing Agricultural University Animal Care and Use Committee (Certification No.: SYXK (Su) 2017-0007).

### 2.2. Animals and Phenotype Measurements

A total of 330 Chinese Suhuai pigs were used in this study, including 247 barrows and 83 females. Experimental pigs were all raised under standard indoor conditions in Huaiyin breeding farm (Huaian, China). Since these pigs were not raised at the same time and the same age (219.10 ± 1.10 day), they could only be slaughtered at the same slaughterhouse in 3 batches in Jinyuan Meat Products Co., Ltd. (Huaian, China) when these pigs reached market weight (80–90 kg). Age, carcass weight, sex, batch, and pedigree were recorded in this study. *Longissimus Dorsi* (LD) muscle samples from the last rib of the left half of the carcasses were collected and used to determine IFC by the Soxhlet extraction method [29]. Ear tissues from the end of the right ear were collected and stored in 75% alcohol for genomic DNA extraction. The BF thicknesses were measured by digital caliper at 3 positions (the shoulder, the last rib and the last lumbar vertebrae) on the right side of the carcass. After that, ABT (ABT = the average values of the results obtained at the 3 positions) was calculated.

### 2.3. Isolation of Genomic DNA and SNPs Genotyping

The genomic DNA was isolated from ear tissue using the standard phenol-chloroform protocol method [30]. The DNA concentration was measured with a Nanodrop 2000 spectrophotometer (Thermo Fisher Scientific Inc, Waltham, MA, USA) and the integrity of the DNA was checked on 1.5% agarose gel with an FR-250 electrophoresis apparatus (Furi Technology Co., Ltd., Shanghai, China). Only high-quality genomic DNAs from all samples were used for subsequent genotyping. The polymerase chain reaction-restriction fragment length polymorphism (PCR-RFLP) method and the improved multiplex ligation detection reaction (iMLDR) technique [31] were used to analyze the polymorphisms of 13 SNPs from the 12 candidate genes (Appendix A). Among these SNPs, 11 were genotyped through iMLDR in one ligation reaction, including *LIPE* (rs328830166), *IGF1* (rs341412920), *IGF1* (rs322131043), *IGF2* (g.3072G > A), *LEP* (rs45431504), *LEPR* (rs45435518), *MC4R* (rs81219178), *RETN* (rs327132149), *RYR1* (rs344435545), *SCD* (rs80912566) and *UBE3C* (rs81329544). Fragments covering these 11 SNPs were amplified using a multiplex of PCR reactions. The multiplex PCR reaction, performed in a 20 μL total volume, contained 1× GC-I buffer (Takara, Dalian, China), 3.0 mM Mg^2+^, 0.3 mM dNTPs, 1 U of Hot-Start Taq DNA polymerase (Takara, Dalian, China), 1 μL of primer mixture, and 20 ng of genomic DNA. The PCR program was as follow: 95 °C for 2 min, 11 cycles (94 °C for 20 s, 65 °C−0.5 °C/cycle for 40 s, and 72 °C for 1 min and 30 s), 24 cycles (94 °C for 20 s, 59 °C for 30 s, and 72 °C for 1 min and 30 s), 72 °C for 2 min, and hold at 4 °C. The purification reaction, performed in a new volume, contained 10 μL of each PCR product, 5U SAP and 2U Exonuclease I at 37 °C for 1 h and at 75 °C for 15 min. The ligation reaction, performed in a 10 μL final volume, contained 1 μL of 10× ligation buffer, 0.25 μL of Taq DNA Ligase (NEB Biotechnology, Beijing, China), 0.4 μL of 5′ ligation primer mixture, 0.4 μL of 3′ ligation primer mixture, 2 μL of purified PCR product mixture and 6 μL of double distilled water (ddH_2_O). The ligation cycling program was 38 cycles (94 °C for 1 min and 56 °C for 4 min), and a hold at 4 °C. A total of 0.5 μL of ligation product was loaded into an ABI3730XL, and the raw data were analyzed using the GeneMapper 4.1 software program (Thermo Fisher Scientific Inc, Waltham, MA, USA). All of the iMLDR primers are presented in Appendix A.

Only the polymorphisms of *FABP3* (rs1110770079) and *PHKG1* (rs697732005) were assessed using the PCR-RFLP method due to the failure of the iMLDR technique. For PCR-RFLP, the sequences for these 2 genes were obtained from the GenBank database (NLM, Bethesda, MD, USA). The primers of these 2 SNPs were designed by Primer Premier 6.0 software (Premier Biosoft International, Palo Alto, CA, USA) (Appendix A). PCR reaction was performed in a final volume of 25 μL, containing 1 μL of the template DNA (50 ng/μL), 1 μL of each primer, 22 μL 1.3 × Taq buffer. Cycling conditions were 98 °C for 2 min; 40 cycles at 98 °C for 10 s, 57 °C for 40 s, 72 °C for 1 min; and a final extension at 72 °C for 5 min. The quality of the PCR product was analyzed by using 1.5 % agarose gel electrophoresis. Genotyping was performed by the TSINGKE Company (Nanjing, China).

### 2.4. Statistical Analysis

Statistical analysis of IFC and ABT of the 330 Suhuai pigs was performed using GraphPad Prism 6 (GraphPad Software Inc, La Jolla, CA, USA) with the quartile method.

Phenotypic correlations of IFC and ABT were calculated using SAS 9.2 software (SAS Institute lnc, Cary, NC, USA). To estimate genetic correlations of IFC and ABT, all the phenotypes were corrected for sex and batch as fixed effects and sample identity (ID) as a random variable using DMU software [32].

In this study, genetic diversity indices were calculated to explore the genetic structure of all SNPs. The minimal number of genotypes used in the analysis was 3, and the total number of animals analyzed here was 327–330. The genotype and allele frequencies were calculated using Microsoft Excel 2013. The heterozygosity (He), homozygosity (Ho) and polymorphic information content (PIC) were estimated using Power Marker V3.0 software [33].

Here, variance analyses were performed using SAS 9.2 software (SAS Institute lnc, Cary, NC, USA). Sex and batch were considered as fixed effects in the models (1) and (2), because sex and batch had significantly effects on both IFC and ABT in variance analyses. We also found that age had significant effects on IFC, and carcass weight had significantly effects on ABT. Thus, age was used as the covariate for IFC in model (1), and carcass weight was used as the covariate for ABT in model (2).

Association analysis of each SNP with IFC and ABT was performed using the PROC GLM procedure of module of SAS 9.2 software. 

For IFC, the following model was applied:
Y_ijklm_ = μ + G_i_ +B_j_ + S_k_ + D_l_ + K_m_ + e_ijklm_,(1)

For ABT, the following model was applied:
Y_ijklm_ = μ + G_i_ +B_j_ + S_k_ + W_l_ + K_m_ + e_ijklm_,(2)

In models (1) and (2), Y_ijlkm_ represents the vector of the phenotypic value of the trait under study, μ in these 2 models is the population mean of IFC and ABT, respectively, G_i_, B_j_ and S_k_ refer to fixed effects of SNPs, batch and sex, respectively, D_l_ represents the covariate of age for IFC, W_l_ is the covariate of carcass weight for ABT, K_m_ is the random variable of kinship matrix using pedigree, and e_ijklm_ represents the random error. All statistical analyses were considered significant at *p* < 0.05.

## 3. Results

### 3.1. Descriptive Statistics for IFC and ABT

Descriptive statistics for IFC and ABT of 330 Suhuai pigs are shown in Figure 2 and Table 1. The means of IFC and ABT were 1.99 ± 0.03% and 26.68 ± 0.28 mm, respectively. The medians of IFC and ABT were 1.86 % and 26.40 mm, respectively. The CVs of IFC and ABT were 31.21% and 19.36%, respectively.

### 3.2. Estimates of Phenotypic and Genetic Correlations of Suhuai Pigs

The phenotypic and genetic correlations between IFC and ABT were calculated. Phenotypic correlation between IFC and ABT was 0.32 (*p* < 0.01) and genetic correlation was 0.34 (*p* < 0.01). These results suggested a moderate correlation between IFC and ABT of Suhuai pigs [34], and thus it was possible to increase IFC without increasing ABT.

### 3.3. Genetic Parameters of These SNPs 

The genotypic and allele frequencies, as well as other genetic parameters (He, Ho and PIC), were estimated for the Suhuai pigs (Table 2). Of these, the PIC values of most SNPs were moderate (0.25 < PIC < 0.5), including the *LIPE* (rs328830166), *IGF2* (g.3072G > A), *MC4R* (rs81219178), *SCD* (rs80912566), *UBE3C* (rs81329544), *FABP3* (rs1110770079), *PHKG1* (rs697732005). The values of He, Ho and PIC of the *LEP* (0.15, 0.85 and 0.14), *LEPR* (0.16, 0.84 and 0.15), *RETN* (0.06, 0.94 and 0.06) and *RYR1* (0.06, 0.94 and 0.06) were low, due to the lack of homozygous mutant genotypes. Moreover, the value of PIC in the *IGF1* (0.09) SNP was low due to the absence of one homozygous genotype.

### 3.4. Association Analysis between SNPs and IFC and ABT

Association analysis results of SNPs with IFC and ABT are shown in Table 3. Of the 13 evaluated SNPs, only the *FABP3* (rs1110770079) SNP was associated with IFC (*p* < 0.05). Among the three genotypes of the *FABP3* gene (rs1110770079), IFC of the GG genotype (2.19 ± 0.09 %) was significantly higher than that of the TT genotype (1.93 ± 0.07 %). In addition, the results revealed a tendency (*p* < 0.1) for differences among the *RYR1* (rs344435545) and *SCD* (rs80912566) genotypes for IFC. However, none of these SNPs was associated with ABT. 

## 4. Discussion

IFC and ABT are important economic traits for meat quality. Research on the genetic basis of these two traits has long been the focus of attention [4,5,6]. Recent studies using RNA sequencing identified 5 genes that affect fat deposition [35]. Genome-wide association studies indicated that the *CTN1* gene is associated with IFC [36]. IFC is a relevant trait for high-quality meat products [37], but high BF thickness leads to a decrease in lean meat percentage [38]. Therefore, this study aimed to find molecular markers that are significantly associated with IFC, but do not change ABT of Suhuai pigs. A total of 330 Suhuai pigs were measured for IFC and ABT. The CVs for IFC and ABT were high, perhaps due to the fact that the Suhuai is a new synthetic hybrid breed of pig, and there was a difference in the age and carcass weight of this population. The mean of IFC was low, but compared with Large White, Suhuai pigs had higher IFC and BF. Phenotypic and genetic correlations between IFC and ABT were moderate. This suggests that it should be possible to increase IFC without changing BF thickness. Suzuki et al. [12] reported that the phenotypic correlation between IFC and ABT was moderate, similar to our results.

In the present study, 13 SNPs from 12 previously reported candidate genes for IFC were genotyped, and these SNPs were polymorphic in Suhuai pigs. Among these SNPs, low polymorphisms of the alleles *IGF1* (rs322131043), *LEP* (rs45431504), *LEPR* (rs45435518), *RETN* (rs327132149) and *RYR1* (rs344435545) were observed. These results indicated that there was a strong selection pressure on several traits that were associated with SNPs in this study, such as growth rate, yield of lean meat, BF thickness and so on. The lower values of PIC and He in the *IGF1* (rs322131043), *LEP* (rs45431504), *LEPR* (rs45435518) and *RETR* (rs327132149) provided additional evidence for the strong selection pressure. For example, the *RYR1* (rs344435545) SNP is a missense mutation that causes a change in protein (Arg^615^→Cys^615^), and this mutation results in muscle dysfunction and finally porcine stress syndrome (PSS) [39]. This disease may cause pigs to produce pale soft exudative meat (PSE) [22]. Therefore, breeders have been working hard to eliminate deleterious alleles in recent decades.

Association analysis of the 13 evaluated SNPs with IFC and ABT was performed. Associations of all SNPs with IFC were not significant, except for the *FABP3* (rs1110770079) SNP. These 13 SNPs did not associate with ABT of Suhuai pigs; the possible reasons were that some of these genes were not causal genes for BF thickness, such as *PHKG1*, *RETN*, *SCD*, and *UBE3C*. The *FABP3* gene is one member of the fatty acid binding protein family, which plays a critical role in intracellular fatty acid transport by binding lipids and regulating metabolic homeostasis [40]. The *FABP3* gene might be responsible for IFC and is often regarded as a candidate gene [13]. Chen et al. [41] found that the *FABP3* (HinfI) SNP was associated with IFC in both Yanan (*p* < 0.001) and DLY (*p* < 0.05) pigs, but did not significantly affect the BF thickness, strikingly similar to the results in our study. Although the rs344435545 SNP in the *RYR1* gene is likely to cause an increase in the incidence of porcine stress syndrome, there is a strong correlation between IFC and *RYR1* gene expression level [42] and as this mutation can cause a decrease of BF thickness, it has been considered as an example of balanced selection [43]. The *SCD* (rs80912566) SNP could affect the fatty acid composition and IFC within the Duroc population [18]. Moreover, the *SCD* gene was identified as a candidate gene related to IFC between pigs with high and low IFC [44]. The result from Henriquez-Rodriguez et al. [45] showed that the *SCD* (rs80912566) SNP was associated with fat composition but not with fat content. Their results were consistent with our study. It is worth noting that the results in Table 3 revealed a tendency (*p* < 0.1) of association between the *RYR1* (rs344435545) and *SCD* (rs80912566) with IFC. We speculate that these two SNPs have small effects on IFC or these may be only in linkage disequilibrium with the causative mutation of IFC.

The *IGF1* gene affects the regulation of adipogenesis. Several results suggested a degree of positive correlation between the *IGF1* gene expression and adipocyte content [15], and within the QTLs that affect IFC and BF thickness. Similar to the *IGF1* gene, the *LEP* and *LEPR* genes also play a role in adipogenesis, and several studies have found that *LEP* (rs45431504) and *LEPR* (rs45435518) are polymorphic and could significantly influence IFC and BF thickness [46,47,48]. The *RETN* gene is located on SSC2, and *RETN* (rs327132149) was significantly associated with the abdominal fat weight, BF thickness and IFC [21,49]. On the contrary, *IGF1* (rs322131043), *LEP* (rs45431504), *LEPR* (rs45435518) and *RETN* (rs327132149) were not significantly associated with IFC and ABT in our study and might be strongly selected for these 3 genes in Suhuai population, resulting in low polymorphisms in these SNPs.

The *LIPE* gene has long been considered as a candidate gene that could affect IFC deposition due to resolving fat [14,50]. Burgos et al. [16] reported that *IGF2* (g.3072G > A) could affect pig carcass traits and IFC in Large White×Landrace populations. On the contrary, Aslan et al. [51] found that the *IGF2* (g.3072G > A) SNP may not affect IFC in Pietrain, Duroc and Large White populations. Supakankul et al. [23] reported that these 2 porcine *UBE3C* polymorphisms (rs81329544 and rs32466023) were associated with IFC and fatty acid composition. The *UBE3C* gene is considered a potential candidate gene for fat deposition in muscle because of its location on SSC18, near the QTLs for IFC and FA composition [52]. The *PHKG1* gene is related to glycolysis potential and could affect pork quality. A previous study indicated that there was a point mutation (rs330928088) in a splice acceptor site of intron 9 in the *PHKG1* gene. This point mutation gave rise to the 32 bp deletion in the open reading frame (ORF) and generated a premature stop codon [20]. However, the *PHKG1* (rs330928088) SNP is not polymorphic and did not cause the 32 bp deletion of Suhuai pigs. The SNP (rs697732005) in front of this *PHKG1* (rs330928088) SNP is polymorphic, although it was not associated with IFC in this study. The *MC4R* gene has only one exon, located in the QTL for ABT on SSC1, and plays an important role in the regulation of energy homeostasis [53]. Several reports found that the *MC4R* gene could be associated with fat mass in humans [54] and IFC in Hu sheep [55]. Lyadskiy et al. [56] found no significant difference between the spinal fat thickness and the *MC4R* (rs81219178) SNP, similar to our results. Nevertheless, there was no significant difference in IFC between the three genotypes of the *MC4R* (rs81219178) in pigs [19]; this result is consistent with our study. The fact that the above-mentioned five SNPs were not significantly associated with IFC and ABT might be due to genetic heterogeneity. After all, Suhuai pig is a hybrid population containing Huai pig (25%) and Large White (75%).

These 13 SNPs of the 12 genes did not associate with ABT in Suhuai pigs. The possible reasons were that some of these genes were not causal genes for BF thickness, and other candidate genes of BF thickness may have heterogeneity between breeds. Finally, the *FABP3* (rs1110770079) SNP was the genetic marker we were looking for, which could improve IFC without increasing BF thickness.

## 5. Conclusions

Our results showed that phenotypic and genetic correlations between IFC and ABT were moderate. These 13 SNPs of the 12 genes were polymorphic in Suhuai pigs. Among them, *FABP3* (rs1110770079) SNP was associated with IFC (*p* < 0.05) but not with ABT, and this confirmed the importance of porcine *FABP3* as a candidate gene for IFC of Suhuai pigs. The cost of genotyping tests is low, and farmers could select favorable genotype (GG) individuals according to the genotyping results for pig breeding. Further studies are necessary to confirm the finding.

## Figures and Tables

**Figure 1 animals-09-00858-f001:**
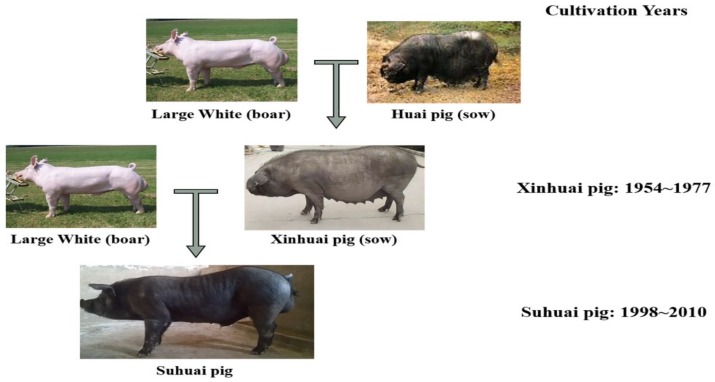
The cultivation process of Suhuai pig.

**Figure 2 animals-09-00858-f002:**
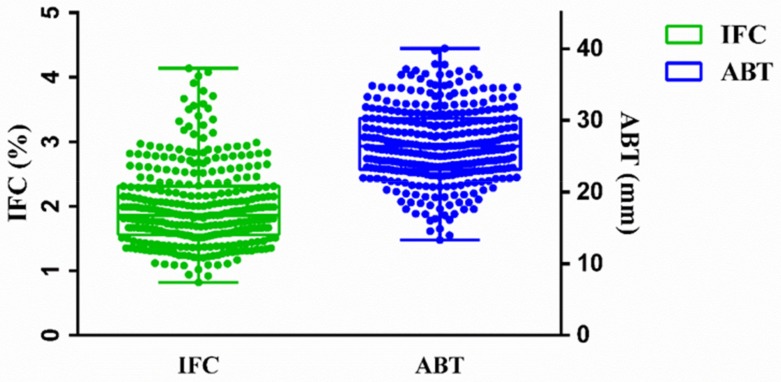
Statistical analysis of IFC and ABT of Suhuai pigs. IFC = intramuscular fat content (green box); ABT = average backfat thickness (blue box); statistical analysis was performed using GraphPad Prism 6 with the quartile method.

**Table 1 animals-09-00858-t001:** Descriptive statistics of the intramuscular fat content (IFC) and average backfat thickness (ABT) in 330 Suhuai pigs.

Trait	Number	Mean ± SE	Max	Min	CV%
IFC	330	1.99 ± 0.03%	4.14%	0.82%	31.21
ABT	330	26.68 ± 0.28 mm	40.02 mm	13.13 mm	19.36

Mean ± SE represents the means with standard errors for diverse genotypes; Max refers to the maximum value of the phenotype; Min refers to the minimum value of the phenotype; CV = coefficient of variation.

**Table 2 animals-09-00858-t002:** Genetic parameters of 13 SNPs in 12 previously reported candidate genes for IFC of Suhuai pigs.

Gene (SNP)	Chromosome	Genotype (Number)	Frequency		HE	HO	PIC
			Genotype	Allele			
*FABP3* rs1110770079	6	GG (82)	0.25	0.44 (G)	0.36	0.64	0.37
		GT (117)	0.36				
		TT (123)	0.38	0.56 (T)			
*LIPE* rs328830166	6	AA (53)	0.16	0.40 (A)	0.48	0.52	0.36
		AG (157)	0.48				
		GG (120)	0.36	0.60 (G)			
*IGF1* rs341412920	5	CC (8)	0.02	0.16 (C)	0.28	0.72	0.24
		CT (90)	0.28				
		TT (228)	0.70	0.84 (T)			
*IGF1* rs322131043	5	AG (32)	0.10	0.05 (A)	0.10	0.90	0.09
		GG (298)	0.90	0.95 (G)			
*IGF2* g.3072G > A	2	AA (62)	0.19	0.42 (A)	0.46	0.54	0.37
		AG (151)	0.46				
		GG (117)	0.35	0.58 (G)			
*LEP* rs45431504	18	CC (1)	0.01	0.08 (C)	0.15	0.85	0.14
		CT (51)	0.15				
		TT (278)	0.84	0.92 (T)			
*LEPR* rs45435518	6	CC (2)	0.01	0.09 (C)	0.16	0.84	0.15
		CT (53)	0.16				
		TT (274)	0.83	0.91 (T)			
*MC4R* rs81219178	1	AA (109)	0.33	0.60 (A)	0.53	0.47	0.37
		AG (175)	0.53				
		GG (45)	0.14	0.40 (G)			
*PHKG1* rs697732005	3	AA (184)	0.57	0.70 (A)	0.26	0.74	0.33
		AG (85)	0.26				
		GG (53)	0.16	0.30 (G)			
*RETN* rs327132149	2	AA (1)	0.00	0.03 (A)	0.06	0.94	0.06
		AG (20)	0.06				
		GG (309)	0.94	0.97 (G)			
*RYR1* rs344435545	6	CC (309)	0.94	0.97 (C)	0.06	0.94	0.06
		CT (20)	0.06				
		TT (1)	0.00	0.03 (T)			
*SCD* rs80912566	14	CC (59)	0.18	0.41 (C)	0.47	0.53	0.37
		CT (154)	0.47				
		TT (117)	0.35	0.59 (T)			
*UBE3C* rs81329544	18	AA (14)	0.04	0.22 (A)	0.35	0.65	0.28
		AG (117)	0.35				
		GG (199)	0.60	0.78 (G)			

HE = heterozygosity; Ho = homozygosity; PIC = polymorphic information content.

**Table 3 animals-09-00858-t003:** Association analysis of 13 SNPs in 12 previously reported candidate genes with the intramuscular fat content (IFC) and average backfat thickness (ABT) of Suhuai pigs.

Gene (SNP)	Genotype	Number	IFC (%)		ABT (mm)	
			Mean ± SE	*p* Value	Mean ± SE	*p* Value
*FABP3* rs1110770079	GG	82	**2.19 ± 0.09 ^a^**	**0.0230**	26.89 ± 0.57	0.6040
	GT	117	**2.13 ± 0.07 ^ab^**		26.18 ± 0.49	
	TT	123	**1.93 ± 0.07 ^b^**		26.49 ± 0.52	
*LIPE* rs328830166	AA	53	2.05 ± 0.08	0.4204	26.82 ± 0.70	0.9068
	AG	157	1.95 ± 0.05		28.14 ± 4.53	
	GG	120	2.03 ± 0.06		27.82 ± 1.03	
*IGF1* rs341412920	CC	8	1.92 ± 0.20	0.5286	26.67 ± 1.61	0.9857
	CT	90	1.94 ± 0.06		26.52 ± 0.51	
	TT	228	2.02 ± 0.04		26.61 ± 0.35	
*IGF1* rs322131043	AG	32	2.00 ± 0.10	0.9981	26.50 ± 0.31	0.5349
	GG	298	2.00 ± 0.04		27.22 ± 1.22	
*IGF2* g.3072G > A	AA	62	2.01 ± 0.07	0.9340	26.50 ± 0.44	0.4130
	AG	151	2.00 ± 0.05		26.59 ± 0.37	
	GG	117	1.98 ± 0.06		26.21 ± 3.20	
*LEP* rs45431504	CT	51	1.98 ± 0.08	0.7460	26.11 ± 0.65	0.3260
	TT	278	2.01 ± 0.04		26.78 ± 0.32	
*LEPR* rs45435518	CT	53	1.88 ± 0.09	0.1070	26.10 ± 0.67	0.4410
	TT	274	2.02 ± 0.04		26.63 ± 0.32	
*MC4R* rs81219178	AA	109	2.04 ± 0.05	0.2504	26.12 ± 0.67	0.7776
	AG	175	1.95 ± 0.05		26.65 ± 0.32	
	GG	45	2.10 ± 0.09		26.82 ± 0.70	
*PHKG1* rs697732005	AA	187	1.98 ± 0.07	0.2221	26.10 ± 0.41	0.1650
	AG	89	2.14 ± 0.08		27.18 ± 0.52	
	GG	36	2.02 ± 0.12		26.74 ± 0.77	
*RETN* rs327132149	AG	20	1.96 ± 0.13	0.7730	27.81 ± 1.03	0.2080
	GG	309	2.00 ± 0.04		26.50 ± 0.31	
*RYR1* rs344435545	CC	309	2.01 ± 0.04	**0.0850**	26.61 ± 0.31	0.9550
	CT	20	1.78 ± 0.13		26.56 ± 1.02	
*SCD* rs80912566	CC	59	2.09 ± 0.08	**0.0510**	27.01 ± 0.46	0.1666
	CT	154	1.88 ± 0.05		26.76 ± 0.48	
	TT	117	2.01 ± 0.06		26.42 ± 0.37	
*UBE3C* rs81329544	AA	14	1.79 ± 0.15	0.1659	27.22 ± 1.22	0.8540
	AG	117	1.95 ± 0.06		26.50 ± 0.44	
	GG	199	2.04 ± 0.05		26.59 ± 0.37	

Mean ± SE represents the means with standard errors for different genotypes; Superscripts with different lowercase letters indicate a significant difference between genotypes (*p* < 0.05).

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
