# Peer review of "Association of Twelve Candidate Gene Polymorphisms with the Intramuscular Fat Content and Average Backfat Thickness of Chinese Suhuai Pigs"

_animals, 2019, doi:10.3390/ani9110858_

Round 1
Reviewer 1 Report
This paper is very interesting and based on solid science. The results are quite impressive if confirmed in further studies. I found two weaknesses in the methodology that need to be addressed regarding the molecular analysis (please see below). Additionally, I have several comments and corrections throughout the manuscript.
The supplementary files do not open. I need a further review if correct files to check the supplementary material.
Title page
This e-mail is quite unusual: 765967561@qq.com (P.L.)
I suggest the authors use institutional e-mail or at least an e-mail with the name.
Additionally, I strongly suggest the authors create an ORCID profile to identify their research.
Simple summary:
Line 17: predominant
Not sure if this is the corrected word.
Lines 20-24
This text is not adequate for simple summary. Only say that you found one marker and explain to a non-scientific readers how this can be important. Write to pig farmers.
Abstract
Line 31: these
Suggest to replace by: Our
You defined two abbreviations. Intramuscular fat (IMF) and average backfat thickness (ABT). In one (ABT) you use the numerical approach average. In the other, you don’t, but always in the paper you refer to IMF as content. So, for standard purposes, why not stipulate that IMF stands for Intramuscular fat content, or use the abbreviation IFC? This will improve the readability of your paper.
Line 32: remove the parentheses.
Line 33: (C.V)
Use CV or C.V.
Line 38: could be regarded
Could be used
Keywords: why to capitalize some of the words and not others. Please standardize
Introduction:
Please put the introduction starting on the second page of the manuscript.
I see here why you used the IMF abbreviation without including the content. If you intend to keep IMF as intramuscular fat I suggest that you use IFC for the intramuscular fat content.
Line 43-44: Meat has been widely consumed being as an important protein source for humans and its consumption in China, especially for pork, is the highest in the world
Poor sentence. Address that the mains kind of meat consumed in chine is pork and put some number regarding the meat consumption (kg per person per year should be included).
Line 52: (lipid droplets)
This is not a correct expression for triglycerides, please remove.
Line 56: Since although the heritability
Poor writing. Please rephrase.
Lines 73-75: Suhuai pig is a new lean-type pig breed, which has 25% Chinese Huai pig blood and 75% 73 Large White blood. Huai pig is one of the North China pig breed, famous for its good meat 74 quality and high forage tolerance
Lack of citations.
Additionally, It should be great a photo of this breed, perhaps in the graphical abstract. Or in the supplementary material. A photo with the genetic backgrounds should be great.
Line 75-76: Historical data states that Huai pig has relatively high IMF content
Lack of citation
Line 76: a commercial pig
Change for: a commercial breed
Lines 76-77: In contrast, Large White pig, a commercial pig which is known for its lean carcass and 76 rapid growth rate, shows lower fat deposition capacity.
Lack of citation
Lines 77-78: Therefore, Suhuai pig occupies those 77 great characteristics from both Chinese Huai pig and Large White pig.
Poor writing, please rephrase.
Line 78: please divided the paragraph before the words: A previous study…
Line 79-80: Based on this, Suhuai 79 pig, as a crossbreed, is a good genetic material for identifying the molecular markers for the 80 IMF content.
This sentence is hard to understand. Please give more details to the readers why the breed is good.
Additional please avoid the use of adjectives such as: great, good. This lack specificity, good relation to what? Correct this throughout the paper.
Lines 81-82: The objective of the present study was to find molecular markers that could 81 improve the IMF content but did not affect the ABT in Suhuai pig.
This is not the objective. And the sentence is wrong. The marker will not improve anything. Please rephrase. You aimed to find markers that are related or indicated a higher IMF content but not ABT, in order to allow farmers to use those markers as a tool for animal selection.
Lines 82-85: In this study, 82 polymorphisms of the 13 above-mentioned SNPs among the 12 previously reported candidate 83 genes for IMF content were genotyped in 330 Suhuai pigs and the associations of different 84 genotypes with the IMF content and average backfat thickness (ABT) were also studied.
This is methodology. Please remove. You can give more details of what you did in the methodology citing the SNP´s polymorphisms in the previous sentence, but this sentence should be removed.
Methodology
I am glad that the authors used a large number of animals. I had reviewed some papers and was concerned about few samples.
Line 98: kinship of each pig
Weird sentence, please rephrase
Line 99: longissimus dorsi
Italic (latin name)
Lines 99-100: Samples of longissimus dorsi (LD) at the last rib and ear samples were 99 collected, then all LD muscle samples were vacuumed and stored at -20 °C.
What you did with the ear samples???
Line 102: spine
The spine is long. Were exactly.
Please define with accuracy the anatomical sites of measurement allowing others to reproduce your work in the future.
Line 103: respectively,
I don´t understand the use of this word here. Perhaps is because the sentence is poor writing.
Line 106: The genomic DNA was isolated from ear tissue by using the standard phenol-chloroform
Please inform the storage condition and give further details of the site in the ear samples.
The phenol-chloroform is a quite aggressive technique that damages the DNA. Who the authors can guarantee the quality of their samples. This should be addressed in the manuscript since more modern techniques for DNA extraction are evaluable that produce better DNA and should be chosen preferably in denser molecular study, such as this one. The phenol method is ok for simple PCR.
Lines 111-112: were used to analyze the polymorphisms of these 13 SNPs from the 12 candidate genes.
This part is confusing. You never cite the genes in the methodology. I think a table here should be great. Be clear regarding the genes and the SNP´s for each gene.
Well, I find a problem here. You use 2 different techniques for different genes. This weakness the paper. Especially since the most important SNP was found in the secondary methodology. Why do you have this failure in the iMLDR? Do you think the results from the different techniques will produce different set of results hard to compare each other?
Lines144-145: It is worth noting that if only one or two individuals in one genotype, individuals with this genotype will be removed and do not participate in subsequent analysis.
Replace this informing the minimal number of genotypes used in the analysis. And inform the total number of animals analyzed here.
Line 154: μ is the population mean
Of what? Define please.
Line 156: is the random variable of kinship
I don´t understand this
Regarding the model, why it was important to include the batch? Do you think this will influence? Did you run the model without the batch? Why you don´t use the covariate of age for ABT model? And why you don´t use carcass weight for IMF content?
Additionally, I think more information regarding the breed should be interesting for readers. When this breed was made? In the present, the breed uses it´s own reproducers (male and female) or the farms are still making a new animals from the 2 original breeds? Some data regarding population (number, number of meat produced, area that is used, ranking the breed among others). This information can be inserted in the introduction or in the methodology.
Results:
Line 162: (1.99±0.03) % and (26.68±0.28) mm,
Please remove the parenthesis.
Please correct the figure position since it is in the middle of the paragraph
Line 164: the figure legend is missing information. In a figure you always must define all the abbreviations. Which kind of statistical? I don´t see any. Please correct the figure and it´s legend.
Lines 167-168: Phenotypic and genetic correlations between the IMF content and ABT were 0.32 and 0.34, respectively
Besides I understood some readers my be confused with this sentence. Please rephrase too: Phenotypic correlation between the IMF content and ABT was 0.32 (p-value) in while the and genetic correlation was 0.34 (p-value).
Line 168: These results suggested a moderate relationship
This is not moderate. This is low correlation. Please provide a reference for moderate relation.
Table 1
The indication is poor. Please use gIMF, gABT, pIMF and pABT for genetic and phenotypic variables. And define these abbreviations in the title of legend of the table.
I don’t understand which kind of data you used here as “genetic correlation”. Please inform.
Line 176: (He, Ho and PIC)
Please define this abbreviations.
Table 2. please define all the abbreviations used in the footnote of the table (not necessary for gene names of course).
Line 189: Of these SNPs…
You used several times this expression. I think this is adequate when you mentioned the SNP´s in the previous sentence. Not here and in other parts. You can use: Of the 13 evaluated SNP´s, only the….
Line 189: SNP significantly associated
SNP was associated with…
Since you are informing the p-value, don´t need the word significantly.
Line 194: between the RYR1 (rs344435545) or SCD (rs80912566) and the IMF content
Replace for: between the RYR1 (rs344435545) and SCD (rs80912566) with the IMF content
Discussion
The discussion is missing recent work in the subject. A quick search in the journal Animals show that several papers are missing in the discussion:
https://www.mdpi.com/2076-2615/9/9/609
https://www.mdpi.com/2076-2615/9/6/313
https://www.mdpi.com/2076-2615/9/7/410
https://www.mdpi.com/2076-2615/9/6/361
https://www.mdpi.com/2076-2615/9/6/314
I recommend to include at least the two first papers in the discussion.
Line 199: take charge of regulating
Poor writing, rephrase
Line 201: aims
Aimed
Line 206: were moderate.
See comment above
Line 206: This suggested that it was still possible
Why the use of the word still here?
I need to revise the discussion after the inclusion of the suggested papers.
Conclusion.
I did not like this conclusion.
The author must give more value to their finding. In the discussion the say that H-FABP is the gene that they were looking for. Here they must comment regarding the cost of their test, the ability to use an ear sample to identify the marker and say that they found one genome that had higher IMF but was not associated with a similar higher ABT, and that farmers now can include this too in the genetic selection of reproducers. Further studies are necessary to confirm the finding.
Author Response
Dear Reviewer,
First of all, thanks for your comments. Your valuable suggestions are very beneficial to us. We have revised our manuscript according to your comments.
Please see the attachment.
Sincerely yours,
Dr Ruihua Huang

Reviewer 2 Report
The study attempts to validate few candidate genes previously reported to be associated with intramuscular fat content in pigs. Not convinced it adds any value to the current body of research. The writing is very poor. Not suitable for publication.
Author Response
Dear Reviewer,
First of all, we are so sorry that our original manuscript did not make you pleasing. Your comments are very beneficial to us.
Point:
The study attempts to validate few candidate genes previously reported to be associated with intramuscular fat content in pigs. Not convinced it adds any value to the current body of research. The writing is very poor. Not suitable for publication.
Response: Thanks for your opinions and guidance, and these valuable and informative comments are very beneficial to us. And we will follow your suggestions and do further research to explore the unique SNPs and genes associated with intramuscular fat content (IFC) in the Suhuai pig population. For this manuscript, we believe that there may be some reference value for pig breeding:
Improving the IFC but not the average backfat thickness (ABT) has become an important objective in modern pig breeding. However, it is difficult to improve the IFC by traditional breeding methods since pigs must be slaughtered to accurately measure the IFC. Marker-assisted selection (MAS) provides a predominant method to improve the IFC of pigs. Although many studies have identified genetic loci associated with the IFC in the past, the causative quantitative trait nucleotides (QTN) were absent. These identified genetic loci were the common variants with small effects due to the linkage disequilibrium (LD) with QTN. Their polymorphisms and associations with the IFC have serious population heterogeneity. These genetic loci are required validation in pig selection, so our research is useful.
Meanwhile, when using SNPs associated with IFC to increase IFC of pig, whether these SNPs will adversely affect the backfat thickness remain unclear. Therefore, our study simultaneously analysed the associations between SNPs with IFC and ABT, which has innovative and application value.
In this study, Suhuai pigs we used are distinctive. Suhuai pig is introduced as a suitable material used for researching the genetic basis of IFC and ABT due to contains both the blood of Huai pig with high IFC and the blood of Large White pig with low IFC. Besides, the large variation of the phenotype were detected within this population.
In summary, we believe that our research has innovation and application value.
We look forward to your further guidance and comments, thank you again.
Sincerely yours,
Dr Ruihua Huang
Reviewer 3 Report
The study, in my opinion, is partially innovative but well conducted and scientifically sound. English needs to be improved.
Materials and methods: Please provide more information about the experiment. Were the pigs raised all at the same time and/or slaughtered the same day or at the same age?
L 96: Please specify the age range of these pigs.
L 193: Authors might indicate P<0.1 when referring to a tendency.
L 200-201: I am not sure that any people in the world prefer to buy pork meat with high IMF. Please provide references for that.
L 205: This sentence is speculative. You did not compare in the present study Large white with Suhuai, therefore I would suggest removing this sentence. Moreover, many factors (such as diet or age at slaughter) might influence IMF or backfat in all breeds.
L 205: Do you have any explanation for this CV?
L 270: This looks like a too-easy explanation for your results. Please improve it arguing more and adding literature.
Conclusion: I would suggest improving the conclusion in order to better enhance your research.
Author Response

(The authors gave the same response as above.)

Round 2
Reviewer 1 Report
The authors extensively correct the entire manuscript according to my suggestions with the inclusion of a new table and additional supplementary material. The figure S1 was a great addition to the manuscript specially to international readers.
The manuscript presents novel and scientific soundness appropriated for publication at the journal.
I strongly suggest to authors in further revision to use the track changes mode of the word software since it allows a better evaluation of the modifications.
Additionally, I suggest the authors move the table S2 and figure S1 from the supplementary material to the main manuscript.
Author Response
Dear Reviewer,
Thanks for your kind comments and valuable suggestions on our manuscript. We have revised our manuscript according to your comments.
Point 1: The manuscript presents novel and scientific soundness appropriated for publication at the journal.
I strongly suggest to authors in further revision to use the track changes mode of the word software since it allows a better evaluation of the modifications.
Additionally, I suggest the authors move the table S2 and figure S1 from the supplementary material to the main manuscript.
Response 1: Thanks for your recognition and suggestions. All of revisions have been clearly highlighted using the "Track Changes" function in Microsoft Word. We have moved the Table S2 and Figure S1 from the supplementary material to the main manuscript. (line 82, page 3, lines 193-197, page 6)
Thank you again for the kind advices.
If you have any questions about this revision, please don’t hesitate to let me know.
Sincerely yours,
Dr Ruihua Huang